# TNFR1 Mediated Apoptosis Is Protective against *Mycobacterium avium* in Mice

**DOI:** 10.3390/microorganisms11030778

**Published:** 2023-03-17

**Authors:** Yuki Shundo, Rintaro On, Takemasa Matsumoto, Hiroshi Ouchi, Masaki Fujita

**Affiliations:** 1Department of Respiratory Medicine, Faculty of Medicine, Fukuoka University, Fukuoka 814-0180, Japan; 2Research Institute for Diseases of the Chest, Graduate School of Medical Sciences, Kyushu University, Fukuoka 812-8582, Japan

**Keywords:** mycobacteriosis, TNF, apoptosis, innate immunity

## Abstract

*Mycobacterium avium* is an intracellular proliferating pathogen that causes chronic refractory respiratory infection. Although apoptosis induced by *M. avium* has been reported in vitro, the role of apoptosis against *M. avium* infection in vivo remains unclear. Here, we investigated the role of apoptosis in mouse models of *M. avium* infection. Tumor necrosis factor receptor-1 knockout mice (TNFR1-KO) andTNFR2-KO micewere used. *M. avium* (1 × 10^7^ cfu/body) was administered intratracheally to mice. Apoptosis in lungs was detected by terminal deoxynucleotidyl transferase mediated dUTP nick end labeling and lung histology as well as cell death detection kits using BAL fluids. TNFR1-KO mice were susceptible to *M. avium* infection compared with TNFR2-KO and wild type mice based on the bacterial number and lung histology. Higher numbers of apoptotic cells were detected in the lungs of TNFR2-KO and wild-type mice were compared with TNFR1-KO mice. The inhalation of Z-VAD-FMK deteriorated *M. avium* infection compared with vehicle-inhaled controls. Overexpression of Iκ-B alpha by adenovirus vector attenuated *M. avium* infection. Our study showed apoptosis had an important role in innate immunity against *M. avium* in mice. The induction of apoptosis in *M. avium*-infected cells might be a new strategy to control *M. avium* infection.

## 1. Introduction

*Mycobacterium avium* causes chronic progressive respiratory infection and is known to be an intracellular proliferating pathogen. *M. avium* sometimes causes disseminated diseases in HIV patients [1,2]. Pulmonary *Mycobacterium avium-intracellulare* complex (MAC) disease is often difficult to control under clinical conditions. The standard treatment regimen for pulmonary MAC disease contains a combination of macrolides (clarithromycin or azithromycin) well known as key drug for MAC disease, rifampicin, and ethambutol. Although several studies reported that macrolide-containing regimens had a 59–92% response rate [3,4,5], relapses are common after medical therapy with guideline-based treatment regimens [6,7]. Recently, addition of amikacin liposome inhalation suspension (ALIS) to standard guideline-based treatment is recommended for patients with refractory MAC disease in the latest ATS/ERS/ESCMID/IDSA clinical practice guidelines [1). The culture conversion rate after initiation of ALIS for 6 months was higher in the ALIS group than in the control group (29.0% vs. 8.9%). Although ALIS is effective medication, ALIS is not recommended for the initial treatment right now. The efficiency of the medical treatment is not satisfied, and novel treatment is necessary. The investigation of host defense mechanisms against *M. avium* infection might lead to new treatments against *M. avium* infection.

Tumor necrosis factor (TNF)-alpha is a pleiotropic cytokine. TNF-alpha demonstrated a critical role in controlling immunological responses in host defense. TNF-alpha is synthesized as a nonglycosylated, transmembrane protein, which undergoes cleavage by a specific metalloproteinase TNF-converting enzyme to form a soluble trimer. There are two receptors bound to TNF-alpha. One is TNFR1 and another is TNFR2. TNFR is located on virtually all cells throughout the body. TNFR bound to TNF-alpha evokes a variety of reactions [8,9,10]. TNF-alpha plays an important role against intracellular proliferating bacilli as innate immunity. These criteria contains *Mycobacterium tuberculosis* or *Listeria monocytogenes* [11,12,13,14,15].

With respect to mycobacteriosis, apoptosis induced by *M. avium* has been reported in vitro [16,17]. TNF-alpha-receptor 1 (TNFR1), but not TNFR2, contains a death domain, which induces apoptosis [18]. TNFR1-knockout (KO) mice are susceptible to *M. avium* infection [19]. Since apoptosis, the process of programmed cell death, is related to many diseases, apoptosis is called attention in an area of extensive research. Apoptosis has a crutucal role in maintaining the delicate balance between cell proliferation and cell death. Apoptosis is also thought to be closely associated with immune mechanisms that protect against mycobacteriosis. However, apoptosis has a limited role in *M. tuberculosis* infection in vivo [20,21] and our understanding of apoptosis in *M. avium* infection is poor. In the present study, we have investigated the role of apoptosis in a mouse model of *M. avium* infection to determine the contribution of apoptosis to host defense against *M. avium*.

## 2. Materials and Methods

### 2.1. Bacteria

Two strains (FM and TM) of clinically-isolated *M. avium* from our hospital were used. All strains were cultured using Middlebrook 7H9 broth with Middlebrook ADC enrichment (Becton, Dickinson and Company, Sparks, MD, USA) at 37 °C. In agar plate, strains were cultured on Middlebrook 7H10 agar with Middlebrook OADC enrichment (Becton, Dickinson and Company, Sparks, MD) at 37 °C for 14 days in 90% humidity. After incubation, colonies were counted as described before [22].

### 2.2. Mice

Eight-week-old female mice were used in this study. TNFR1-KO [6], TNFR2-KO [15], and lpr (Fas-KO) gld (FasL-KO) mice were from the Jackson Lab (Bar Harbor, ME). Perforin-deficient mice (perforin-KO) were kindly provided by Dr. H. Hengartner [23]. Wild type C57BL/6 mice were used as controls. Nuclear factor (NF)-κB deficient mice were a kind gift from Dr. Kubota, Kyushu University [24]. Because mice with a targeted disruption of the p50 subunit of NF-κB [25] were on a BALB/c background, wild type BALB/c were used as controls. In sterile food and water in an environmentally controlled room, these mice were bred.

### 2.3. Animal Model of M. avium Infection

*M. avium* (1 × 10^7^ CFU/head) in 50 µL of sterile saline was administered intratracheally. These procedure was performed by tracheotomy of the mice under anesthesia as described elsewhere [26,27]. As a control, 50 µL of saline was injected. Mice were sacrificed on days 3, 21, or 60 after infection and the lungs were dissected and homogenized. The homogenates were inoculated in Middlebrook 7H10 agar plates to count the number of colonies. Z-VAD FMK, a caspase inhibitor, was nebulized from day 1 to 7 after *M. avium* administration to wild type mice. An adenovirus vector encoding an inhibitor of NF-κB (Iκ-B) alpha (AdIκ-B alpha) and LacZ adenovirus (AdLacZ) was constructed as reported elsewhere [28]. To construct these recombinant viruses, an adenovirus type 5 lacking the E1 region was used. The recombinant adenoviruses were grown in 293 cells. Then virus were purified by centrifugation on CsCl_2_ gradients. A dose of 3 × 10^8^ plaque forming units/mouse AdLacZ or 50 µL vehicle (saline containing 10% glycerol) was injected into the left lung of mice. This study was approved by the Institutional Animal Care and Use Committee of Kyushu University. Also the study was approved by the Institutional Animal Care and Use Committee (IACUC) of Fukuoka University. The study was followed the Guidelines for Animal Experimentation, Fukuoka University. The IACUC are charged with protecting the safety and welfare of animals used in research at or in conjunction with Fukuoka University. 

### 2.4. Lung Histology and Morphometry

After sacrifice, the lungs were fixed with 10% formalin for 24 h and embedded in paraffin. Then hematoxylin and eosin (H-E) or Ziehl-Neelsen (Z-N) staining were done. As previously described with some modifications [28,29], morphological evaluation was also performed. The average score of lung injury on a scale of zero to eight were graded as follows: grade 0, normal lung; grade 1, minimal lung injury; grade 2, moderate thickening of walls; grade 4, increased lung injury; grade 6, severe distortion of the lung structure; and grade 8, total obliteration of the lung.

### 2.5. Bronchoalveolar Lavage (BAL)

BAL was performed on day 21 after *M. avium* infection as described elsewhere [30]. The trachea was exposed and intubated, and the lungs were lavaged five times with 1 mL of Ca^2+^ and Mg^2+^ free phosphate buffered saline (PBS) at 4 °C. The cells were counted and stained with modified Wright’s stain (DiffQuik; American Scientific Products, McGas Park, IL) for differential counts.

### 2.6. Macrophage Isolation and Infection by M. avium

Bacterial clearance from macrophages was performed as described elsewhere [31] with some modifications. Briefly, at 3 days after inoculation of mice with 2 mL of 3% thioglycollate medium (Becton, Dickinson and Company, Franklin Lakes, NJ), peritoneal macrophages were obtained. Macrophages were incubated in RPMI-1640 medium without antibiotics. *M. avium* was added into the culture medium at a multiplicity of infection (moi) 10. Then, macrophages were collected. After washed twice in PBS, the macrophages were lysed with the addition of sterilized water. Lysed medium was inoculated into Middlebrook 7H10 agar plates. The bacterial number was counted after incubation at 37 °C.

### 2.7. Western Blotting

Lungs were obtained from mice at day 21 after *M. avium* infection, homogenized, and then analyzed by immunoblotting using a 1:2000 dilution of mouse anti-caspase-3 antibody purchased from Santa-Cruz Biotechnology Inc. (Santa Cruz, CA) as described elsewhere [28]. Finally, the blots were reacted with a chemiluminescent detection substrate by using ECL Western Blotting Substrate kit (Thermo Fisher Scientific, Tokyo, Japan).

### 2.8. Apoptosis Detection

Apoptosis in lungs was detected by terminal deoxynucleotidyl transferase mediated dUTP nick end labelling (TUNEL) using lung histology as described elsewhere [32]. A Cell Death Detection ELISA PLUS kit (Roche Diagnostics GmbH, Penzberg, Germany) was used for BAL fluids according to the manufacturer’s protocol. This kit determines cytoplasmic histone-associated DNA fragments after induced cell death.

### 2.9. Statistical Analysis

Data were expressed as the mean ± standard error (SE). The Mann-Whitney U-test was used to compare two groups. A *p*-value < 0.05 was considered to indicate statistical significance. Statistical analyses were performed using StatView 5.0 (SAS Institute, Cary, NC, USA).

## 3. Results

### 3.1. M. avium Inoculation

Although TNFR1 have a death domain, a signaling element that induces apoptosis, TNFR2 does not have a death domain. Therefore, TNFR1-KO mice could not induce apoptosis, while TNFR2-KO mice could induce apoptosis. The inoculation of *M. avium* caused severe pathological changes in TNFR1-KO mice but not in wild type mice or TNFR2-KO mice (Figure 1A–D). Ziehl-Neelsen staining indicated the proliferation of acid-fast bacilli was involved in the pathology observed in the lungs of TNFR1-KO mice (Figure 1E,F). In this experiment, TNFR1-KO mice were proved to be susceptible for *M. avium* infection. Then, we investigated other apoptosis-gene deleted mice. Compared with wild type mice, Fas and FasL-deficient mice also demonstrated severe pathologic changes, especially, granuloma formation. Morphometric analyses performed for semi-quantitative analyses demonstrated severe pathologic involvement in TNFR1-KO mice. Lpr and gld mice had moderate pathologic involvement between wild type and TNFR2-KO mice (Figure 2A). Bacterial growth in the lungs of mice 21 days after *M. avium* inoculation was examined. Bacterial growth in the lungs was similar to pathological changes observed by H-E staining (Figure 2B). This experiment demonstrated apoptosis could contribute to immune-defense mechanism against *M. avium* infection Then bacterial growth in peritoneal macrophages after in vitro *M. avium* inoculation was determined (Figure 3). There was no difference in the proliferation of *M. avium* in macrophages from TNFR1- or TNFR2-KO mice. Because there was no difference in bacterial growth in vitro, mechanisms other than internal cell killing might contribute to the susceptibility of mice to *M. avium*.

### 3.2. Apoptosis in the Lungs of Mice Infected with M. avium

Histology of TUNEL staining on day 21 after *M. avium* inoculation was investigated to detect apoptosis in lungs. There were more TUNEL positive cells (apoptotic cells) in the lungs of wild type mice than in TNFR1-KO mice (Figure 4A,B). To determine whether numbers of apoptotic cells were decreased in the lungs by a method other than TUNEL, we used a Cell Death Detection ELISA kit for the relative quantification of histone-complexed DNA fragments. We performed a cell death detection assay using BAL fluids on day 21 after *M. avium* inoculation (Figure 4C,D). Alveolar macrophages were the predominant inflammatory cells in BAL fluids. Wild type and TNFR2-KO mice had high levels of histone-complexed DNA fragments, indicators of apoptosis, in the BAL fluid. In addition, we examined caspase-3, apoptosis marker, after *M. avium* infection. Caspase-3 was detected in wild type mice but faintly in TNFR1-KO mice. An inverse relationship between cell death and pathological changes was observed.

We used NF-κB-deficient mice to determine how survival signals affected *M. avium* infection in vivo. Lung histology of H-E-stained tissues on day 60 after *M. avium* inoculation in NF-kB-deficient mice was investigated. In this experiment, the background of NF-kB-deficient mice is BALB/c, wild type of BALB/c mice were used. BALB/c mice demonstrated similar lung histology compared to C57BL/6. There was no difference of pathogenesis of background. Compared with wild type mice, NF-kB-deficient mice did not develop severe pathologic changes (Figure 5). This experiment demonstrated that lack of survival signals did not affect the immune system against *M. avium* infection.

### 3.3. Apoptosis Modification

The effect of the inhibition or promotion of apoptosis on *M. avium* proliferation in vivo was investigated. First, the nebulization of Z-VAD-FMK, an inhibitor of apoptosis, was instilled. Z-VAD nebulization was previously shown to inhibit apoptosis [33]. Histology and bacterial growth demonstrated the progression *M. avium* infection after Z-VAD-FMK inhalation compared with controls (Figure 6). The experiment proved that inhibition of apoptosis lead to deterioration of *M. avium* infection. The overexpression of Ik-B alpha, a promoter of apoptosis, attenuated the pathologic changes and inhibited bacterial growth induced by *M. avium*. In this experiment, perforin-KO mice were used, because they are susceptible to *M. avium* infection (data not shown). In addition, the overexpression of Ik-B alpha inhibited bacterial growth in the lungs and induced high levels of histone-complexed DNA fragments in the BAL fluid (Figure 7). These data indicated that the induction of apoptosis led to the inhibition of bacterial growth and attenuate lung injury by *M. avium* infection.

## 4. Discussion

In the present study, we investigated the mechanism involved in protection against *M. avium*, especially focusing on apoptosis. TNFR1-KO and Fas/FasL-deficient mice were susceptible to *M. avium* infection compared with TNFR2-KO mice and wild type mice in vivo. TNFR1-KO and Fas/FasL-deficient mice (gld/lpr) lacked the signals to induce apoptosis and therefore developed less apoptosis in the lungs or BAL fluid after *M. avium* inoculation. Interestingly, NF-κB deficient mice did not demonstrate severe lung inflammation after *M. avium* inoculation. Thus, apoptosis in vivo, probably in alveolar macrophages, has a critical role in host defense against *M. avium*.

TNF-α binds to receptors of TNFR1 and TNFR2. TNFR is presented on virtually all cells. TNF-α evokes a variety of inflammatory reactions. Because of the crucial role of TNF-α in inflammation, infection and immunity, the role of TNF-α and its receptors has been vigorously investigated. As a result, TNFR1 plays a major role in host immunity against pathogens. However, the role of TNFR2 remains uncertain. TNFR1-KO mice, but not TNFR2 KO mice, contain a death domain [18,34]. If apoptosis contributed to the death of mice, then TNFR2-KO mice should have shown severe lung disease. However, the pathogenesis of TNFR2-KO mice was similar to that of wild type mice. In contrast, TNFR1-KO mice, lacks apoptosis signals, demonstrated severe lung disease after *M. avium* inoculation. Several studies reported that TNFR1-KO mice were susceptible to *M. avium* infection [35] as shown in this study. Furthermore, the apoptosis of macrophages was reported to contribute to protective immunity against mycobacterial infection [16,17,36]. Interestingly Fratazzi and colleagues reported that non-infected macrophages phagocytosed infected macrophages to eliminate *M. avium* [37]. Taken together, we consider that apoptosis in macrophages has a crucial role in protective immunity against *M. avium* infection.

The relationship between the TNF receptors and bacterial infection have already been reported. For the primary response to *Listeria monocytogenes*, TNF-α and TNFR1 have a crucial role [13]. With regard to extracellular proliferating bacteria such as *Escherichia coli*, TNFR deficiency contributed to compromise bacterial killing, however, TNFR deficiency did not suppress inflammation [38]. TNF-α signaling through TNFR1 has an important role for host defense in both *Streptococcus pneumoniae* [39], and *Klebsiella pneumoniae* infection [40]. Although the role of TNFR1 for innate immunity against bacteria has been established, the role of TNFR2 has not been clarified. In this study, the difference between TNFR1 and TNFR2 against *M. avium* was established. In future, the role of TNFR2 against other bacteria will be more clarified.

To confirm further the effect of apoptosis on *M. avium* infection, mice inhaled Z-VAD-FMK, a caspase inhibitor. Inhalation of Z-VAD-FMK could lead to inhibition of apoptosis [33]. The inhalation of Z-VAD-FMK deteriorated *M. avium* infection compared with vehicle-inhaled controls. This experiment proved that inhibition of apoptosis lead to progression of *M. avium*-infection severity. Then, the promotion of apoptosis was investigated by the overexpression of Ik-B alpha. NF-κB is usually kept inactive in the cytoplasm through an association with an endogenous inhibitor protein of the Iκ-B (inhibitor of NF-κB) family [41]. The overexpression of Iκ-B alpha could lead to promotion of apoptosis. As a result, the overexpression of Iκ-B alpha induced promotion of apoptosis and resulted in the attenuation of *M. avium* infection compared with the AdLacZ-inhaled control. Either the induction or inhibition of apoptosis clearly affected *M. avium* infection in vivo. Thus, apoptosis was considered to have an important role in protective immunity against *M. avium* in mice. Apoptosis induction could lead to attenuation of *M. avium* infection, and inhibition of apoptosis could lead to progression of *M. avium* infection.

The phagocytosis ability was not different among wild type mice, TNFR1-KO mice, and TNFR2-KO mice, because the bacterial number in lysates obtained immediately after *M. avium* inoculation was similar. Interestingly, the bacterial growth in macrophages from TNFR1-KO mice was similar to that in wild type mice. Bermudez and colleagues reported that several strains of *M. avium* used to infect human macrophages for 5 days triggered 28–46% higher levels of apoptosis than that observed with uninfected macrophages at the same time points [42]. Apoptosis also prevented the release of intracellular components and the spread of mycobacterial infection by sequestering pathogens within apoptotic bodies [37]. As mentioned previously, apoptosis of host macrophages might be an important defense mechanism in mycobacterial infections to prevent the spread of infection. Although the attachment of cells to a culture dish may have compromised the phagocytosis of apoptotic cells, it is likely that macrophage functions in vitro are completely different from those in vivo. Out data demonstrated that internal cell killing by phagocytosis is insufficient as a defense mechanism against *M. avium*. To eliminate *M. avium*, a two-step process might be necessary. First, cells (mainly macrophages) infected with *M. avium* undergo apoptosis and are then phagocytosed in the second step.

In conclusion, the data of our study indicate that apoptosis might have an important role in innate immunity against *M. avium* in mice. The induction of apoptosis in *M. avium*-infected cells, mainly macrophages, might be a new strategy for controlling *M. avium* infection. Currently, the major limitations for the effective therapy of non-tuberculous mycobacteriosis are the absence of specific antimicrobial agents with low toxicity and good in vivo activity against non-tuberculous mycobacteria [1]. For example, compared with against *M. tuberculosis,* most first-line anti-TB drugs have 10 to 100 times less in vitro activity against *M. avium* isolates. It is difficult to treat *M. avium* infection because there is no efficient treatment. Current treatment, a combination of rifampicin, ethambutol, macrolides, and ALIS, is not sufficient for cure. This study indicates that the induction of apoptosis in *M. avium*-infected cells, mainly macrophages, might be a new strategy to control *M. avium* infection in vivo as previously reported in an in vitro study by Fratazii et al. [43]. We should seek the more appropriate method for controlling apoptosis in *M. avium* infection in future.

## Figures and Tables

**Figure 1 microorganisms-11-00778-f001:**
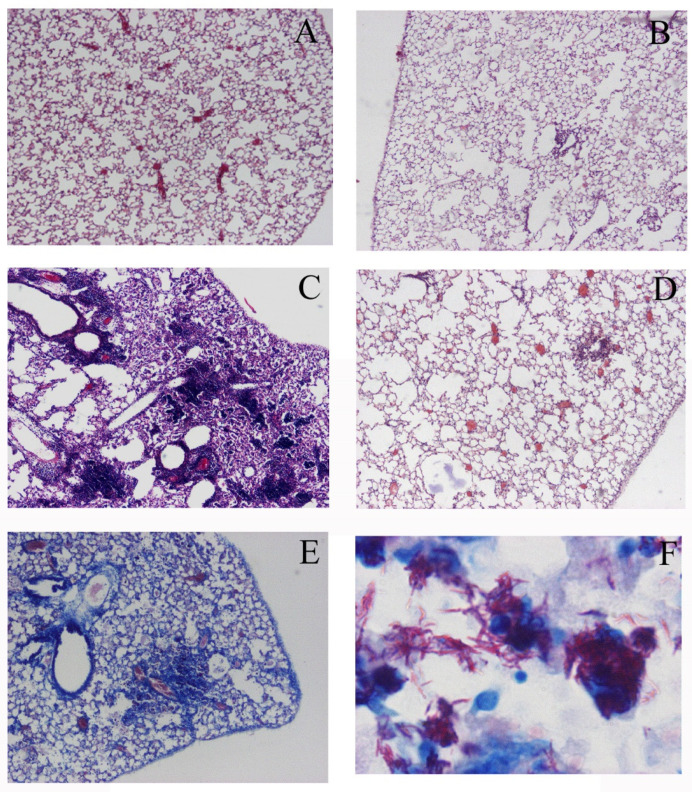
Histology of lungs from mice 60 days after *M. avium* administration. (**A**) Wild mice (C57BL/6) administered saline, (**B**) wild mice administered *M. avium*, (**C**) TNFR1-KO mice administered *M. avium*, and (**D**) TNFR2-KO mice administered *M. avium*. The marked inflammatory cell infiltrate in the lungs from TNFR1-KO mice was noted. (**A**–**D**) panels are H-E stained and viewed at the same magnification. Original magnification ×40. Ziehl-Neelsen staining revealed the proliferation of *M. avium* in lungs (**E**), (**F**) magnification ×1000. Three experiments were done and demonstrated similar results.

**Figure 2 microorganisms-11-00778-f002:**
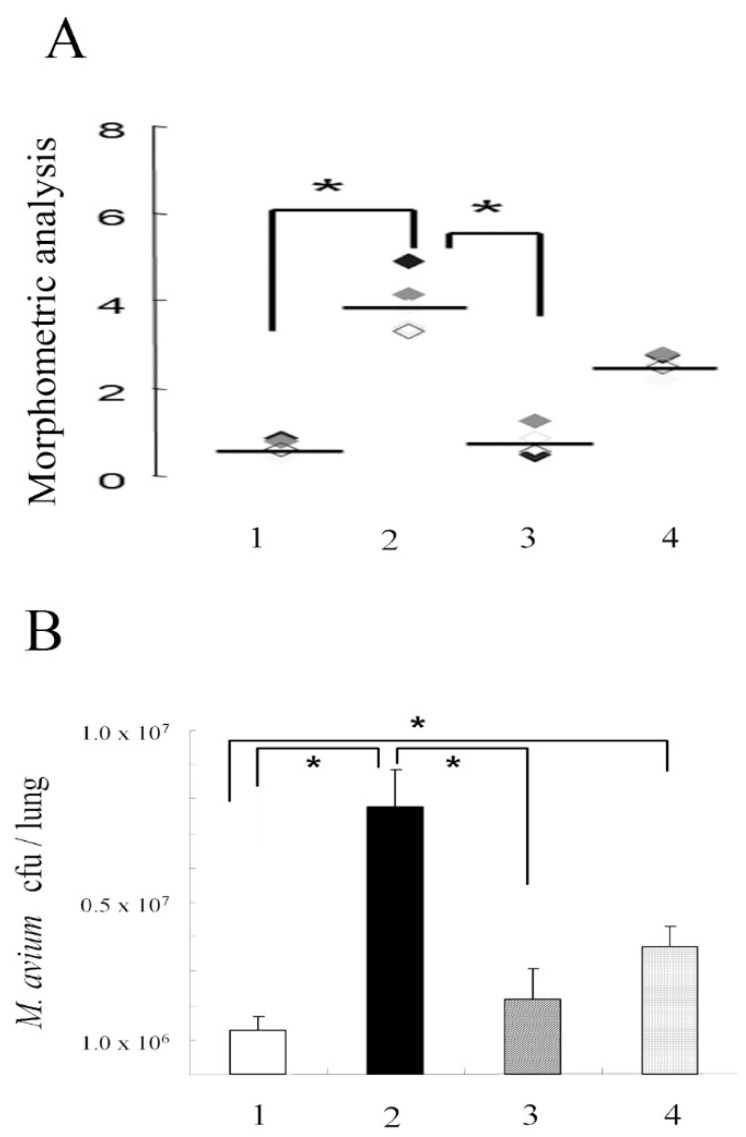
The effect of apoptosis related gene destruction on *M. avium* administration. (**A**) Morphometric analysis of the lung. Morphometric analysis was carried out using the lungs from mice 60 days after *M. avium* administration. 1: Wild-type mice of C57BL/6, 2: TNFR1-KO mice, 3: TNFR2-KO mice, 4: lpr mice (Fas-KO mice). Different color squares represents individual mice for each group. (**B**) Bacterial growth of *M. avium* in the lungs of mice. Bacterial growth in the lungs of mice was measured at day 21 after *M. avium* inoculation. 1: Wild-type mice of C57BL/6, 2: TNFR1-KO mice, 3: TNFR2-KO mice, 4: lpr mice (Fas-KO mice). In both experiments, each group consisted of five mice. * Indicates significant differences (*p* < 0.05).

**Figure 3 microorganisms-11-00778-f003:**
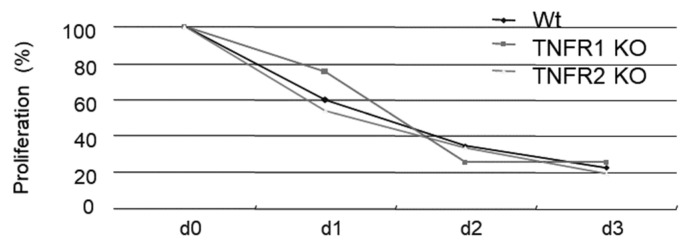
Macrophages from wild mice (C57BL/6), TNFR1-KO, or TNFR2-KO mice were infected with *M. avium* at an moi of 10. At days 1, 2, and 3, bacterial growth was counted. Bacterial growth is shown as a log scale compared with day 0. Each group consisted of five mice. There was no significant difference in bacterial growth between groups.

**Figure 4 microorganisms-11-00778-f004:**
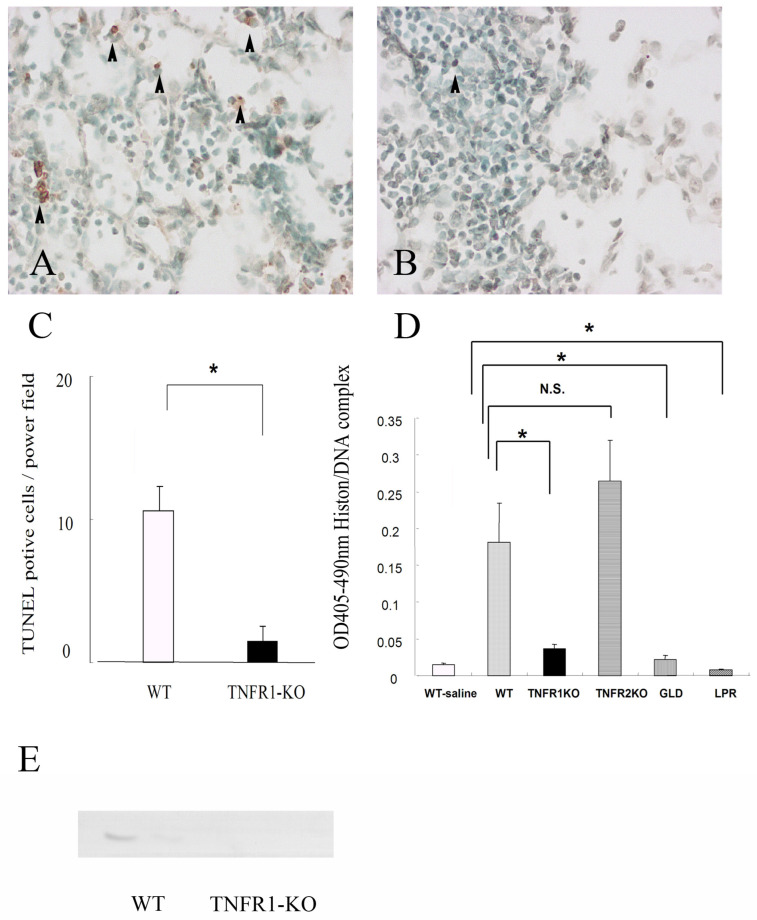
The effect of TNFR1 deficiency on terminal deoxynucleotidyl transferase mediated dUTP nick end labelling (TUNEL) staining in lung tissues after *M. avium* infection. (**A**) TUNEL positive cells are present in lung tissues from wild mice (C57BL/6) 21 days after *M. avium* infection. (**B**) There were fewer positive TUNEL signals (arrowheads) in TNFR1-KO mice. Original magnification, ×250. Arrowheads indicate TUNEL positive cell. (**C**) The effect of TNFR1 deficiency on the number of terminal deoxynucleotidyl transferase mediated dUTP nick end labelling (TUNEL) positive cells in *M. avium* infection in mice. Data are shown as the mean (SEM) obtained from five mice (* *p* < 0.01). (**D**) The effect of apoptosis-related gene deficiency on histone/DNA complexes in BAL fluids at day 21 after *M. avium* infection. WT-saline: wild-type (C57BL/6) mice administered saline, WT: wild type mice (C57BL/6) administered *M. avium*, TNFR1KO: TNFR1-KO mice administered *M. avium*, TNFR2KO: TNFR2-KO mice administered *M. avium*, lpr: lpr (Fas-KO) administered *M. avium*, gld: gld (FasL-KO) mice administered *M. avium*. High titers of histone/DNA complexes were observed in wild-type mice administered *M. avium*. Data are shown as the mean (SEM) obtained from five mice (* *p* < 0.01). N.S. indicates no significance. (**E**) The expression of caspase-3 was measured by western blotting. Caspase-3 was strongly expressed at day 21 after *M. avium* infection in wild type mice (C57BL/6), but not in TNFR1-KO mice.

**Figure 5 microorganisms-11-00778-f005:**
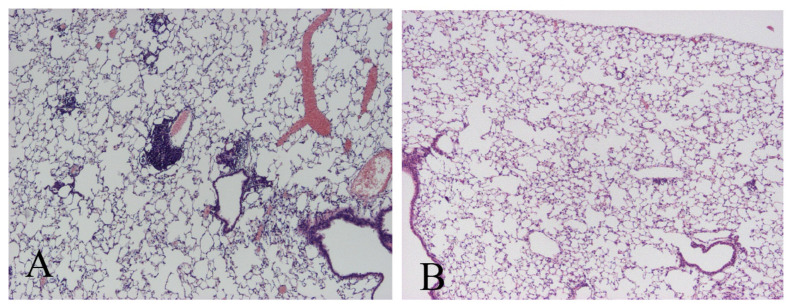
Histology of lungs from mice 60 days after *M. avium* administration. (**A**) Wild mice (BALB/c) administered *M. avium,* (**B**) NF-kB deficient mice administered *M. avium*. Note the marked inflammatory cell infiltrate in the lungs from wild mice compared with NF-κB deficient mice. Original magnification ×40. Three experiments were done and demonstrated similar results.

**Figure 6 microorganisms-11-00778-f006:**
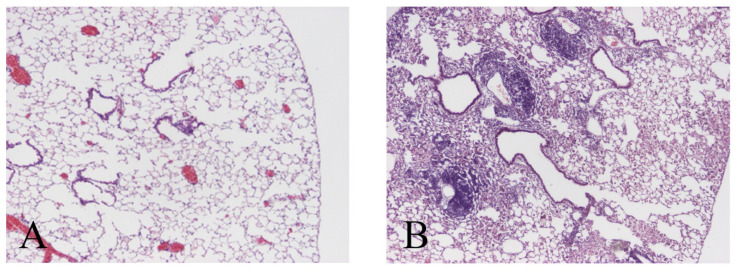
Histology of lungs from mice 60 days after *M. avium* administration. (**A**) Wild mice (C57BL/6) administered *M. avium* followed by inhalation of vehicle (1% DMSO in saline). (**B**) Wild mice administered *M. avium* followed by inhalation of Z-VAD FMK (1 mg/mL). The marked inflammatory cell infiltrate in the lungs from wild mice after Z-VAD FMK inhalation was noted. Original magnification ×40. Three experiments were done and demonstrated similar results.

**Figure 7 microorganisms-11-00778-f007:**
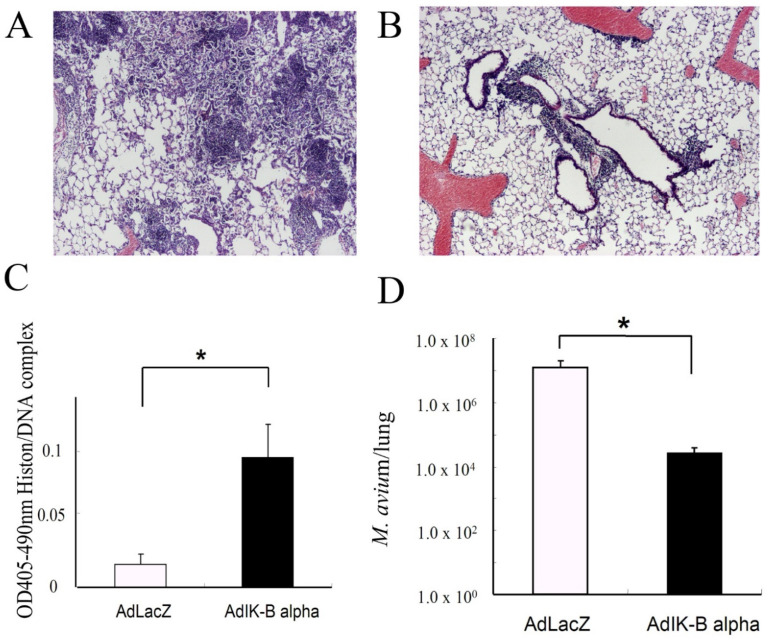
Histology of lungs from mice 60 days after *M. avium* administration. (**A**) Perforin-KO mice administered *M. avium* followed by instillation of AdLacZ. (**B**) Perforin-KO mice administered *M. avium* followed by instillation of AdIκ-B. The reduced inflammatory cell infiltrate in the lungs from Perforin-KO mice with AdIκ-B alpha was noted. H-E staining, original magnification ×40. Three experiments were done and demonstrated similar results. (**C**) The effect of apoptosis-related gene deficiency on histone/DNA complexes in BAL fluids at day 21 after *M. avium* infection. AdLacZ: mice administered AdLacZ as a control, AdIκ-B alpha: mice administered AdIκ-B alpha. High titers of histone/DNA complexes were observed in mice administered AdIκ-B alpha. Data are shown as the mean (SEM) obtained from five mice (* *p* < 0.01). (**D**) Bacterial growth of *M. avium* in the lungs of mice at day 21 after *M. avium* inoculation. Each group consisted of five mice. * Indicates significant differences (*p* < 0.05). AdLacZ: mice administered AdLacZ as a control, AdIκ-B alpha: mice administered AdIκ-B alpha.

## Data Availability

Not applicable.

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
