# Peer review of "TNFR1 Mediated Apoptosis Is Protective against Mycobacterium avium in Mice"

_microorganisms, 2023, doi:10.3390/microorganisms11030778_

Round 1
Reviewer 1 Report
The authors showed that TNFR1-KO mice were susceptible to M. avium infection compared with TNFR2-KO and wild type mice based on the bacterial number and lung histology. In addition, the inhalation of Z-VAD-FMK (a caspase inhibitor) deteriorated M. avium infection compared with vehicle-inhaled controls. Overexpression of Ik-B alpha (a promoter of apoptosis) by adenovirus vector attenuated M. avium infection. Finally, they concluded that apoptosis had an important role in innate immunity against M. avium in mice. These results are really interesting. However, some modifications are required to be accepted.
Major
1.
Both TNFR1 and TNFR2 activate NF-kB. Therefore, the reviewer believes that evaluation of NF-kB activation in infected lung tissues of TNFR1-KO and TNFR2-KO mice is necessary. In other words, the reviewer would like the authors to clarify the role of NF-kB in this study; in Figure 5, the reviewer asks that the authors could show the results of CFU or Ziehl-Neelsen staining as well as pathological findings. Although the authors state ''Interestingly, NF-kB deficient mice were not susceptible to M. avium inoculation.'', P12, L280-L281, there is a lack of data to support this. 
2.
As the authors mentioned, studies using TNFR1-KO mice infected with M. avium have been conducted so far. The reviewer asks that the authors would discuss the differences between the previous studies and the results of the current study.
3.
The reviewer asks the authors to explain why Perforin-KO mice were used in the experiments in Figure 7.
4.
Several studies in which bacteria other than M. avium were used to infect TNFR1-KO and TNFR2-KO mice. The reviewer asks that the authors would discuss the differences between those reports and the results of the present study.
Minor
1.
Please provide a citation for ''TNFR1-knockout (KO) mice are susceptible to M. avium infection.'' in P2, L54-L55.
2.
If possible, the reviewer asks that the authors could describe the survival curves of TNFR1-KO mice, TNFR2-KO mice, and WT mice during M. avium infection.
3.
With regard to ''Compared with wild type mice, Fas and FasL-deficient mice also demonstrated severe pathologic changes, especially, granuloma formation.'', P4, L159-L161, the reviewer asks that the authors would describe enlarged pathology findings (TNFR1-KO, TNFR2-KO and Fas and FasL-deficient mice) to show differences in granuloma formation.
4.
As for ''Histology and bacterial growth demonstrated the progression M. avium infection after Z-VAD-FMK inhalation compared with controls (Figure 6).'', P9, L242-L244, the reviewer asks that the authors would indicate the results of CFU or Ziehl-Neelsen staining, if the authors describe ''bacterial growth''.
Author Response
Reivewer 1
Comments and Suggestions for Authors
The authors showed that TNFR1-KO mice were susceptible to M. avium infection compared with TNFR2-KO and wild type mice based on the bacterial number and lung histology. In addition, the inhalation of Z-VAD-FMK (a caspase inhibitor) deteriorated M. avium infection compared with vehicle-inhaled controls. Overexpression of Ik-B alpha (a promoter of apoptosis) by adenovirus vector attenuated M. avium infection. Finally, they concluded that apoptosis had an important role in innate immunity against M. avium in mice. These results are really interesting. However, some modifications are required to be accepted.
I thank the reviewer for the positive feedback on our manuscript.
Major
1.
Both TNFR1 and TNFR2 activate NF-kB. Therefore, the reviewer believes that evaluation of NF-kB activation in infected lung tissues of TNFR1-KO and TNFR2-KO mice is necessary. In other words, the reviewer would like the authors to clarify the role of NF-kB in this study; in Figure 5, the reviewer asks that the authors could show the results of CFU or Ziehl-Neelsen staining as well as pathological findings. Although the authors state ''Interestingly, NF-kB deficient mice were not susceptible to M. avium inoculation.'', P12, L280-L281, there is a lack of data to support this. 
Thank you for pointing out. I am afraid that I have never done the measurement of CFU or Z-N staining. I agree with your opinion, therefore, I changed the sentence, NF-kB deficient mice did not demonstrate severe inflammation after M. avium inoculation.
2.
As the authors mentioned, studies using TNFR1-KO mice infected with M. avium have been conducted so far. The reviewer asks that the authors would discuss the differences between the previous studies and the results of the current study.
Thank you for pointing out Thank you for pointing out. Although the role of TNFR1 for innate immunity has been established, the role of TNFR2 has not been clarified. In this study, the difference between TNFR1 and TNFR2 against M. avium was established. The difference was considered to attribute to the apoptosis.
3.
The reviewer asks the authors to explain why Perforin-KO mice were used in the experiments in Figure 7.
Thank you for pointing out. In this experiment in Figure 7. I expected the inhibition effect using AdIK-B alpha, therefore, I need the susceptible model for M. avium inoculation rather than TNF-signal gene-deletion model. Since perforin-KO mice was susceptible for M. avium (now, preparation for submission), I used these mice in this experiment. Also perforin-KO mice is more available than lpr/gld mice in my institute.
4.
Several studies in which bacteria other than M. avium were used to infect TNFR1-KO and TNFR2-KO mice. The reviewer asks that the authors would discuss the differences between those reports and the results of the present study.
Thank you for pointing out. In my previous paper, I described concerning other bacteria other than M. avium (Cytokine 2008).
Several reports provide evidence of the relationship between the TNF receptors and bacterial infection. The requirement of TNF-alpha and TNFR1 in the primary response to L. monocytogenes has been established. In the model of extracellular proliferating bacteria such as Escherichia coli, the absence of TNFR, and the lack of both TNFR1 and TNFR2, compromised bacterial killing, but did not suppress inflammation. TNFR signaling is essential to the effective pulmonary host defense against E. coli. With regard to Streptococcus pneumoniae, TNF-alpha signaling through TNFR1 is crucial in resistance, as well as in Klebsiella pneumoniae infection. Although the role of TNFR1 for innate immunity against bacteria has been established, the role of TNFR2 has not been clarified. In this study, the difference between TNFR1 and TNFR2 against M. avium was established. In future, the role of TNFR2 against other bacteria will be more clarified.
Minor
1.
Please provide a citation for ''TNFR1-knockout (KO) mice are susceptible to M. avium infection.'' in P2, L54-L55.
Thank you pointing out. I add to the reference.
Ehlers S, Benini J, Kutsch S, Endres R, Rietschel ET, Pfeffer K Fatal granuloma necrosis without exacerbated mycobacterial growth in tumor necrosis factor receptor p55 gene-deficient mice intravenously infected with Mycobacterium avium.. Infect Immun. 1999 Jul;67(7):3571-3579.
2.
If possible, the reviewer asks that the authors could describe the survival curves of TNFR1-KO mice, TNFR2-KO mice, and WT mice during M. avium infection.
Thank you for pointing out. All mice were surviving within 60 days. I add the sentence in Results section.
3.
With regard to ''Compared with wild type mice, Fas and FasL-deficient mice also demonstrated severe pathologic changes, especially, granuloma formation.'', P4, L159-L161, the reviewer asks that the authors would describe enlarged pathology findings (TNFR1-KO, TNFR2-KO and Fas and FasL-deficient mice) to show differences in granuloma formation.
Thank you for pointing out. I am afraid that I could not find out the difference among TNFR1-KO, Fas-KO and FasL-KO mice.
4.
As for ''Histology and bacterial growth demonstrated the progression M. avium infection after Z-VAD-FMK inhalation compared with controls (Figure 6).'', P9, L242-L244, the reviewer asks that the authors would indicate the results of CFU or Ziehl-Neelsen staining, if the authors describe ''bacterial growth''.
Thank you for pointing out. I am afraid that I could not perform CFU count and Z-N staining.
Reviewer 2 Report
1. The chemiluminescent detection substrate has to be indicated in MM section
2. Fig. 1 - changes should be indicated in the legend, as well as arrow marked clearly
Author Response
- The chemiluminescent detection substrate has to be indicated in MM section
Thank you for pointing out. ECL Western Blotting Substrate were used.
- 1 - changes should be indicated in the legend, as well as arrow marked clearly
Thank you for pointing out. I draw arrow clearly in figure 4 and change the legend.